# Evaluation of dietary selenium methionine levels and their effects on growth performance, antioxidant status, and meat quality of intensively reared juvenile *Hypophthalmichthys molitrix*

**Maida Mushtaq**[1]*, **Mahroze Fatima**[1], **Syed Zakir Hussain Shah**[2], **Noor Khan**[3], **Saima Naveed**[4], **Muhammad Khan**[4]

**1** Department of Fisheries and Wildlife, Faculty of Fisheries and Wildlife, University of Veterinary and Animal Sciences, Lahore, Pakistan, **2** Department of Zoology, University of Gujrat, Gujrat, Pakistan, **3** Department of Zoology, University of Punjab, Punjab, Pakistan, **4** Department of Animal Nutrition, University of Veterinary and Animal Sciences, Lahore, Pakistan

☯ These authors contributed equally to this work.
* maida.ch17@gmail.com

## Abstract

The objective of this study was to optimize the organic selenium (Se) requirements of intensively reared silver carp (*Hypophthalmichthys molitrix*). A total of n = 300 juveniles silver carp 11.40±0.52 cm long, and average weighing 25.28±0.18 grams were randomly assigned to 15 aquaria (20 fish/100L aquaria) and subjected to five different dietary Se levels in a completely randomized design. The diets were pelleted supplemented with exogenous Se methionine @ 0.0, 0.3, 0.6, 0.9 and 1.2 mg/kg of the diet. The fourteen days of aquaria acclimatization was given to fish and then an 84-day feeding trial was conducted. The group supplemented with 0.9 mg/kg Se had greater feed intake, gain in length, body weight %, and specific growth rate with a better feed conversion ratio as compared to those fed on the rest of the dietary levels or control (P<0.05). The deposition of Se was greater in the liver, and kidneys of the fishes fed on diets containing 0.9 and 1.2 mg Se levels than in the rest of the treatments (P<0.05). However, dietary Se levels had no effects on the bioaccumulation of Se in muscle tissues (P>0.05). The proximate analysis showed that dry matter, crude protein, and fat contents of meat were not changed (P>0.05) by dietary treatments. Similarly, values of TBARS, RBCs, Hb, and blood glucose contents were similar (P>0.05) across the treatments. However, the concentration of WBCs, HCT, and MCHC was greater in those groups fed on 0.9 and 1.2 Se levels than in those fed on 0.6, 0.3, and 0.0 Se levels respectively (P<0.05). The activities of ALT, AST, and ALP were lowered in the 0.9 mg Se supplemented fishes compared with the rest of the treatments (P<0.05). The SOD, catalases, and GPx levels for muscle, liver, and whole body were greater (P<0.05) in the Se-supplemented groups than in the control. These outcomes indicated that up to 0.9 mg/kg inclusion of methionine-based Se in the diet of juvenile silver carp improved the

**Data Availability Statement:** All relevant data are within the paper and its Supporting Information files.

**Funding:** The authors received no specific funding for this work.

**Competing interests:** The authors have declared that no competing interests exist.

growth performance, feed conversion ratio, organs Se enrichment, and antioxidant status without any compromise on meat quality.

## Introduction

The silver carp is an important freshwater aquaculture commodity in Pakistan, China, and India. Silver carp can attain a maximum length of 140 cm and weigh up to 50 kg [1]. It has relatively higher market demand because of delicious meat, more lean carcass percentage, and better shelf life. In addition, silver carp has better adaptation to confinement with faster growth rates, and better feed conversion [2] which makes it a viable specie to fit in modern intensive rearing systems. Although this system increased aquaculture production efficiency around the world, high density, which is common practice, caused oxidative stress by producing Reactive oxygen species (ROS) like peroxides of hydrogen, superoxide anion, and nitric oxide [3, 4]. Several studies showed that higher levels of ROS production can lead to disturbances in fish physiological responses, and antioxidant status leading to poor performance, and higher rates of morbidity, and mortality [5, 6]. Therefore, practical techniques that could boost this species' antioxidative status and growth efficiency have drawn more interest.

Endogenous enzymes (superoxide dismutase, catalase, glutathione peroxidase) and exogenous antioxidant substances like Se make up the antioxidant defense system [7]. Selenium is the key trace element required to smooth the growth and metabolism of animals including fish [8]. Research is established in different animal species that dietary Se deficiency led to lower growth performance, exudative diathesis, nutritional muscular dystrophy, and reproductive issues in different animal species [9–13]. The Se toxicity due to higher dosage led to digestive upset and down-regulation of the immune system of the fish [14]. Selenium combined with cysteine amino acid to form selenocysteine (Se-Cys), a well-known 21 amino acid containing cofactor that exists in 25 different forms in the animal body and is an integral part of the glutathione peroxidase (GPx) system that protects cells by catalyzing hydrogen peroxide and lipid peroxide reduction [15]. Several studies evaluated organic and inorganic dietary Se sources in hybrid striped bass [16], common carp [17], crucian carp [18], Nile tilapia [19], and common barbel [20]. According to these studies, organic sources have higher digestibility and bioavailability with better tissue accumulation as compared to inorganic sources.

The organic sources are yeast fermented (Se yeast) and chelated micronutrients (Se methionine). Se methionine has relatively higher bioavailability, digestibility, and organ enrichment properties than Se yeast [18]. Moreover, Se methionine is comparatively less toxic than inorganic sources [14]. In a recent study [1], we compared the sodium selenite, Se yeast, and Se methionine sources with two levels (0.5 and 1 mg/kg of the diet) in silver carp. Feeding 0.5 mg Se methionine/kg of the diet, resulted in stronger silver carp with better performance. According to our knowledge, this was the first study in which Se requirements for the fingerlings of silver carp were investigated. Moreover, dietary selenium requirements differ according to the fish type, stage of growth, and rearing environment. Therefore, the current study aimed to evaluate the Se methionine in different doses and their influence on growth performance, feed intake, meat chemical composition, Se bioaccumulation, blood biochemistry, and antioxidant response of juvenile silver carp.

## Materials and methods

### Ethical approval, experimental design, diets, and fish husbandry

The protocols and procedures of this study were approved by the animal use and animal care committee of the University of Veterinary and Animal Sciences, Lahore, Pakistan (DR/175,

**Table 1. Diets formulation and levels of Se methionine supplementation on a dry basis.**

| Feed Ingredients | Selenium levels[1] | | | | |
|---|---|---|---|---|---|
| | **0.0** | **0.3** | **0.6** | **0.9** | **1.2** |
| Fish Meal (g/kg) | 200.00 | 200.00 | 200.00 | 200.00 | 200.00 |
| Soybean Meal (g/kg) | 200.00 | 200.00 | 200.00 | 200.00 | 200.00 |
| Maize Gluten 60 (g/kg) | 200.00 | 200.00 | 200.00 | 200.00 | 200.00 |
| Wheat Flour (g/kg) | 145.00 | 145.00 | 145.00 | 145.00 | 145.00 |
| Rice Polish (g/kg) | 150.00 | 150.00 | 150.00 | 150.00 | 150.00 |
| Fish Oil (g/kg) | 60.00 | 60.00 | 60.00 | 60.00 | 60.00 |
| Vitamin Premix (g/kg)[2] | 20.00 | 20.00 | 20.00 | 20.00 | 20.00 |
| Mineral Mixture (g/kg)[3] | 20.00 | 20.00 | 20.00 | 20.00 | 20.00 |
| Choline Chloride (g/kg) | 5.00 | 5.00 | 5.00 | 5.00 | 5.00 |
| Selenium Methionine (mg/kg) | 0.00 | 0.30 | 0.60 | 0.90 | 1.20 |

[1]Selenium levels = basal diet containing supplementation of selenium methionine @ 0 mg/kg (0.0), 0.3 mg/kg (0.3), 0.6 mg/kg (0.6), 0.9 mg/kg, (0.9) and 1.2 mg/kg (1.2).
[2]Locally customized and each kg of vitamin premix contains: Vitamin A 15 M.I.U, Vitamin D3 [3]M.I.U, Nicotinic acid 25000mg, Vitamin B1 5000mg, Vitamin E 6000IU, Vitamin B2 6000mg, Vitamin K3 4000mg, Vitamin B6 4000mg, Folic acid 750mg, Vitamin B12 9000mg, Vitamin C 15000mg, Calcium Pentothenate 10000mg.
[3]Locally customized and each kg of Mineral mixture contains: $MgSO_4.7H_2O$ 153mg, $CoCl.6H_2O$ 0.0816mg, NaCl 51mg, $AlCl_3.6H_2O$ 0.255mg $CuSo_4.5H_2O$ 210.67mg, $FeSo_4.H_2O$ 100.67mg, $MnSo_4.5H_2O$ 116.67mg, $ZnSO_4.7 H_2O$ 121.33mg and Cellulose 65mg.

05-04-2022). A total of n = 300 juveniles silver carp 11.40±0.52 cm long, and average weighing 25.28±0.18 grams were randomly assigned to 15 aquaria (20 fish/100L aquaria) and subjected to five different dietary Se levels in a completely randomized design. The diets were pelleted and contained 0.0, 0.3, 0.6, 0.9 and 1.2 mg/kg exogenous Se methionine supplementation. The feed ingredients along with Selenomethionine (Selisseo® 2% Se, Europe) were procured from a local market for this experiment. To produce the pellets (3 mm diameter), feed ingredients were ground through a 0.05 mm sieve (KENWOOD, AT284), batched according to basal diet ingredients inclusion levels, supplemented with Se levels, then each batch was thoroughly mixed (mixture; KENWOOD, AT283), dough by adding water, and then pelleted (meat mincer; ANEX, AG 3060). The fresh pellets were subjected to shade drying to achieve moisture up to 10% and then stored in airtight zipper bags. Feed ingredients, as well as diets, were analyzed by using the proximate method according to AOAC, [21]. The fourteen days ponds acclimatization was given to fish and then an 84-day feeding trial was conducted. The offer feed and orts were managed according to Mushtaq, [1] to calculate the feed intake. During the experimental period, each aquarium's dissolved oxygen level, temperature, and pH were maintained @ 5.8–7.3 mg/L, 24.9–28.7˚C, and 7.4–8.6. respectively. Moreover, each aquarium was subjected to water change on daily basis. The feed formulation, chemical composition, Se methionine levels, and dietary concentrations on a dry basis are given in Tables 1 & 2.

**Table 2. Diets chemical composition on a dry basis.**

| Nutrients | Selenium levels[1] | | | | |
|---|---|---|---|---|---|
| | **0.0** | **0.3** | **0.6** | **0.9** | **1.2** |
| Dry matter (g/kg) | 898.70 | 898.70 | 898.70 | 898.70 | 898.70 |
| Crude protein (g/kg) | 316.20 | 316.20 | 316.20 | 316.20 | 316.20 |
| Crude fat (g/kg) | 123.20 | 123.20 | 123.20 | 123.20 | 123.20 |
| Crude fiber (g/kg) | 75.40 | 75.40 | 75.40 | 75.40 | 75.40 |
| Ash (g/kg) | 92.30 | 92.30 | 92.30 | 92.30 | 92.30 |
| Selenium (mg/kg) | 0.15 | 0.30 | 0.58 | 0.89 | 1.18 |

[1]Selenium levels = basal diet containing supplementation of selenium methionine @ 0 mg/kg (0.0), 0.3 mg/kg (0.3), 0.6 mg/kg (0.6), 0.9 mg/kg, (0.9) and 1.2 mg/kg (1.2).

## Growth parameters

The body length and weight of each fish were measured just before the start of the feeding trial, biweekly, and on the day of termination of the experiment. Eqs 1–4 were used to calculate the different growth parameters.

$$Weight\ gain(g) = Final\ weight(g) - Initial\ weight(g) \qquad (1)$$

$$Weight\ gain(\%) = [(Final\ weight - initial\ weight)/Initial\ weight(g)] \times 100 \qquad (2)$$

$$Feed\ conversion\ ratio = dry\ feed\ intake(g)/Wet\ weight\ gain(g) \qquad (3)$$

$$SGR = [(Final\ weight - Initial\ weight)/days\ of\ growth\ trial] \times 100 \qquad (4)$$

## Sample collection and laboratory analysis

After the growth experiment, seventeen fish from each tank were randomly selected, weighted, and anesthetized in the laboratory with tricane methanesulphate (MS-222) at 150 mg L1, as prescribed by Mushtaq, [1]. Ten fish were blood sampled using plain tuberculin syringes by puncturing the caudal vasculature and collected blood samples were immediately centrifuged at 3000 x *g* for 15 minutes, and harvested serum was stored at -20˚C for further analysis. Furthermore, blood samples were collected via caudal vein puncture with EDTA-coated tuberculin syringes and immediately analyzed for hematological parameters with an automated hematology analyzer (MEK6550). The concentration of glucose in the blood was determined using commercial kits (21503, Biosystems, Barcelona, Spain). Five fish were dissected for organ collection and biological indices, and the other three were homogenized in a meat mincer (ANEX, AG 3060) to estimate the chemical composition of the meat. Organs (kidney, liver, gills, and pancreas) and meat samples were collected and stored at -20 $^{0}$C in labeled plastic airtight zipper bags for further analysis. To calculate the dry matter contents, the collected feed and meat samples were dried for 48 hours at 55 $^{0}$C in a forced-air oven. Using a Foss grinder, dried feed and meat samples were ground and passed through a 1 mm sieve (CT 293 Cyclotec, Denmark). The crude protein (Method 976.06) and fat contents of ground feed and meat samples were determined using the AOAC, [21] standard procedures (Soxtec procedure, Tecator, Hoganas, Sweden; method 920.29). To determine the ash content of ground meat and feed samples, they were ignited in a muffle furnace at 620 $^{0}$C for 3 hours. The concentration of selenium in experimental diets and fish tissues was determined using atomic absorption spectroscopy (Hitachi ZA3000, Japan). In brief, 250 *g* of each tissue/feed sample was mineralized at 85˚C for 4 hours in a closed vessel heating block system with 4 mL of 70% HNO3 (329756406; Sigma-Aldrich) and 2 mL of 35% H2O2 (3587191; SAFC) (Mars5, CEM, USA). The Thiobarbituric acid reactive substances (TBARS) and antioxidant enzyme activities estimation were performed according to the given detailed methodology in our study Mushtaq, [1].

## Statistical analysis

The data of the current experiment was analyzed by using the General Linear Model procedure of SAS (Online version) with diets as a fixed factor/independent variable. Means were separated by using the Tukey test and declared significant at P<0.05 (Tukey, 1991). The dietary Se level for the maximum response for performance variables (WG, and SGR) showed $R^2$ significant and was predicted by using the following linear broken-line regression model of SAS

(Online version).

$$Y = L + U \times (R-X) \times 1$$

Where: Y = dependent variable, L = theoretical maximum, R = requirement, X = independent variable, I = 1 (if X < R) or I = 0 (if X > R), U = rate constant.

## Results

### Growth performance

During the growth trial, the fish survival rate was recorded at 100% for the 0.0, 0.6, and 0.9, while, at 99 and 98% for the 0.3 and 1.2 Se levels respectively. The means, stander error of means, and *P linear* values for growth performance of silver carp fed different Se methionine levels are given in Table 3. The initial weight, as well as length, were similar across the treatments (P>0.05). The other growth parameters like feed intake, final weight, weight gain, final length, feed conversion ratio, and specific growth rate were significantly influenced (P<0.05) by Se levels. The feed intake was higher in supplemented groups and the group supplemented with 1.2 mg Se had a greater intake as compared to the rest of the treatments (P<0.05). Overall growth performance was improved (P<0.05) with Se supplementation. The growth performance parameters like final weight, final length, BW gain, and specific growth rate were significantly higher (P<0.05) in the 0.9 mg/kg Se supplemented group as compared with the rest of the supplemented groups. In addition, the broken line results for weight gain % and SGR showed 0.9 mg/kg as an optimum dietary Se methionine level (Figs 1 and 2). Similarly, those fish supplemented with 0.9 mg Se had better feed efficiency than those fed either on other supplemented levels.

### Tissues bioaccumulation of selenium

The means values with stander error of means for organ enrichment with Se of silver carp fed on different levels of selenium methionine are given in Table 4. The bioaccumulation of Se in the liver and kidney was influenced *(P<0.05)* by different dietary Se levels. The deposition of Se was higher for 0.9 and 1.2 Se levels respectively than in the rest of the treatments. However, Se enrichment in muscle was not influenced by dietary Se levels *(P>0.05)*.

**Table 3. Growth performance of juvenile silver carp fed on different levels of selenium methionine.**

| Parameters | Selenium levels[1] | | | | | SEM | P-linear |
|---|---|---|---|---|---|---|---|
| | 0.0 | 0.3 | 0.6 | 0.9 | 1.2 | | |
| Feed intake (g) | 29.23[a] | 31.32[b] | 31.32[b] | 31.23[b] | 32.10[c] | 0.00 | < .0001 |
| Initial weight (g) | 25.31 | 25.26 | 25.27 | 25.27 | 25.29 | 0.06 | 0.38 |
| Final weight (g) | 36.13[a] | 41.00[b] | 41.71[b] | 49.81[d] | 44.16[c] | 0.43 | < .0001 |
| Weight gain (g) | 10.82[a] | 15.74[b] | 16.44[b] | 24.54[d] | 18.87[c] | 0.31 | < .0001 |
| Weight gain (%) | 42.77[a] | 62.34[b] | 65.08[c] | 97.14[e] | 74.67[d] | 1.77 | < .0001 |
| Feed conversion ratio | 2.78[a] | 2.10[a] | 1.93[b] | 1.29[d] | 1.73[c] | 0.05 | < .0001 |
| Initial length (cm) | 11.49 | 11.39 | 11.37 | 11.39 | 11.36 | 0.03 | 0.26 |
| Final length (cm) | 18.19[a] | 18.97[a] | 19.62[b] | 23.08[d] | 20.49[c] | 0.10 | < .0001 |
| Specific Growth rate (g) | 0.12[a] | 0.18[b] | 0.18[b] | 0.28[d] | 0.21[c] | 0.00 | < .0001 |
| Survival rate (%) | 100.00 | 99.00 | 100.00 | 100.00 | 98.00 | | |

[1]Selenium levels = basal diet containing supplementation of selenium methionine @ 0 mg/kg (0.0), 0.3 mg/kg (0.3), 0.6 mg/kg (0.6), 0.9 mg/kg, (0.9) and 1.2 mg/kg (1.2).

[a-e] superscripts within the same row differ significantly at P<0.05 while, cells within the same row containing shared subscript have no statistically significant difference (P>0.05).

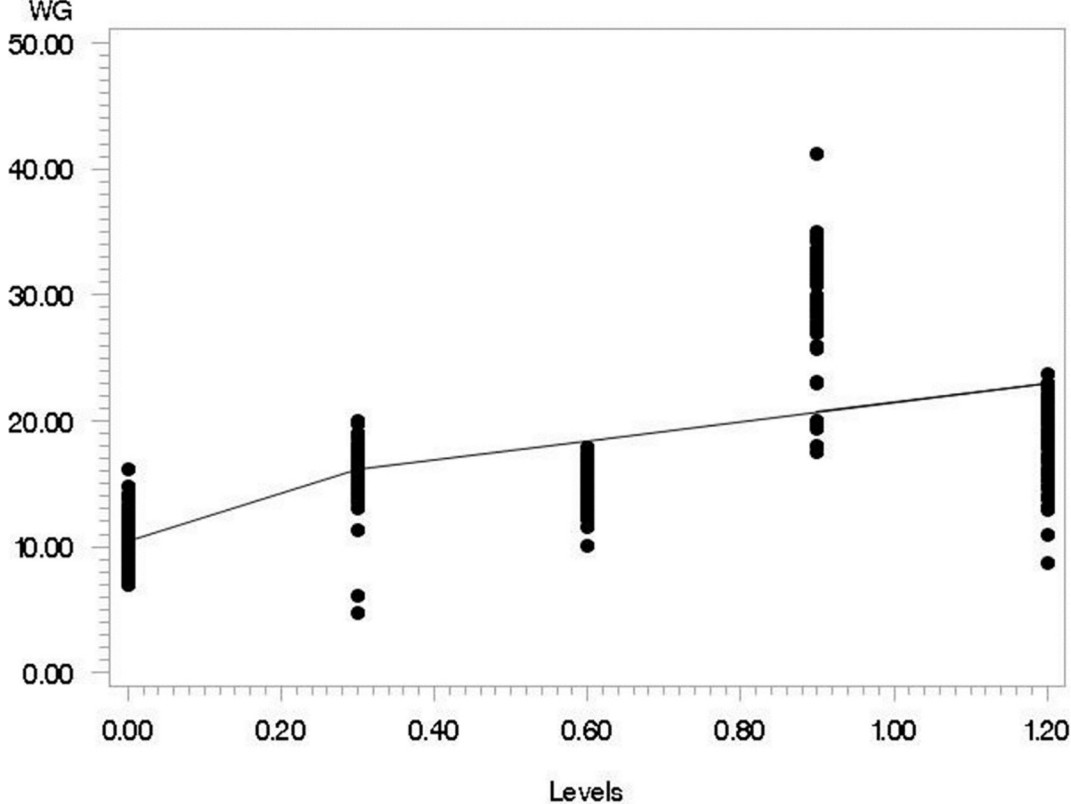

**Fig 1. The fitted broken-line plot of % weight gain of *Hypophthalmichthys molitrix* fed on different levels of selenium methionine.** Liner broken-line fitted model for selenium levels; Y = 60.12–0.13 (102-X) × I, I = 1 (if X<102) or I = 0 (if X>102), *P* < *.006*, $R^2$ = 0.30, the breakpoint occurred at 96±2.12.

## Meat chemical composition

The means, stander error of means, and significant differences among treatments for chemical analysis of the silver carp meat are given in Table 5. The changes in dry contents, fat, ash, and protein contents were non-significant (P>0.05) among the treatments.

## Hematology and serum biochemistry

The results for blood biochemical properties of silver carp fed on different levels of selenium methionine are given in Table 6. The means of WBCs, HCT, MCHC, ALT, AST, and ALP were significantly changed (P<0.05) among treatment groups. However, the means for RBCs, Hb, and blood glucose concentration were similar (P>0.05) among the treatments. The concentration of WBCs, HCT, and MCHC was greater in the 0.9 and 1.2 Se supplemented groups than in 0.3, 0.6, and 0.0 Se fed groups respectively. The activities of ALT, AST, and ALP were lowered in the 0.9 Se supplemented group compared with the rest of the treatments.

## TBARS and enzyme assay

The results of TBARS, as well as antioxidant enzymes of silver carp, fed different Se levels are given in Table 7. The changes in TBARS were non-significant (P>0.05) among the treatments. However, means of catalases, SOD, and glutathione peroxidase levels for muscle, liver, and

## Specific Growth rate

**Fig 2. The fitted broken-line plot of SGR (specific growth rate) of *Hypophthalmichthys molitrix* fed on different levels of selenium methionine.** Liner broken-line fitted model for selenium levels; $Y = 0.1921 - 0.22 (100 - X) \times I$, $I = 1$ (if $X < 100$) or $I = 0$ (if $X > 100$), $P < .002$, $R^2 = 0.60$, the breakpoint occurred at $0.27 \pm 0.02$.

whole body were influenced (P<0.05) by dietary Se levels. The catalases and SOD were higher for the supplemented groups. Similarly, glutathione peroxidase was highest in 0.9 and 1.2 Se supplemented diet than in other levels.

## Discussion

The current experiment results revealed that the optimal dietary Se level was essential for efficient growth performance and to smoothen the physiological function of silver carp. The

**Table 4. Bioaccumulation of selenium in different organs of juvenile silver carp fed on different levels of selenium methionine.**

| Parameters | Selenium levels[1] | | | | | SEM | *P-linear* |
|---|---|---|---|---|---|---|---|
| | 0.0 | 0.3 | 0.6 | 0.9 | 1.2 | | |
| Liver (µg/g) | 0.40[a] | 0.43[a] | 0.45[a] | 0.57[b] | 0.55[b] | 0.01 | 0.022 |
| Kidney (µg/g) | 0.27[a] | 0.29[a] | 0.33[b] | 0.37[d] | 0.37[d] | 0.02 | 0.020 |
| Muscle (µg/g) | 0.19 | 0.22 | 0.24 | 0.29 | 0.275 | 0.03 | 0.261 |

[1]Selenium levels = basal diet containing supplementation of selenium methionine @ 0 mg/kg (0.0), 0.3 mg/kg (0.3), 0.6 mg/kg (0.6), 0.9 mg/kg, (0.9) and 1.2 mg/kg (1.2).
[a-e] superscripts within the same row differ significantly at *P< .05* while, cells within the same row containing shared subscript have no statistically significant difference (*P>0.05*).

**Table 5. Meat chemical composition of juvenile silver carp fed on different levels of selenium methionine.**

| Parameters | Selenium levels[1] | | | | | SEM | *P-linear* |
|---|---|---|---|---|---|---|---|
| | **0.0** | **0.3** | **0.6** | **0.9** | **1.2** | | |
| Dry matter (%) | 44.35 | 44.25 | 43.93 | 43.45 | 43.70 | 0.26 | 0.7899 |
| Crude protein (%) | 16.49 | 16.47 | 16.30 | 16.23 | 16.61 | 0.13 | 0.9021 |
| Crude fat (%) | 7.33 | 7.17 | 7.16 | 7.38 | 7.39 | 0.11 | 0.3251 |
| Ash (%) | 5.76 | 5.57 | 5.45 | 5.56 | 5.34 | 0.08 | 0.1036 |

[1]Selenium levels = basal diet containing supplementation of selenium methionine @ 0 mg/kg (0.0), 0.3 mg/kg (0.3), 0.6 mg/kg (0.6), 0.9 mg/kg, (0.9) and 1.2 mg/kg (1.2).

survival rate suggests that selenium had an impact on the survival of the silver carp. The growth performance parameters like weight gain, and SGR showed significant differences among treatments. It is documented that higher dietary Se levels suppressed the growth performance as 1.2 mg/kg Se supplementation reverse the growth performance in this study [20]. The higher Se levels phenomenon regarding poor SGR is well established in aquatic species, like *Epinephelus malabaricus* [4] and *Acanthopagrus schlegelii* [22]. In the current experiment, the results of the broken-line regression for SGR and weight gain % showed that the optimal exogenous organic dietary Se which is methionine based required 0.9 mg/kg for silver carp. The results regarding different fish species Se requirements are not consistent some reported higher [4, 23], and others reported lower [24]. This difference might be due to the several incomputable variations in fish species, environmental differences [25], biochemical properties of different Se sources [26], and mathematical models used to find the Se requirement [27]. Therefore, we can infer that this research would provide the baseline regarding the Se

**Table 6. Blood biochemical properties of juvenile silver carp fed on different levels of selenium methionine.**

| Parameters | Selenium levels[1] | | | | | SEM | *P-linear* |
|---|---|---|---|---|---|---|---|
| | **0.0** | **0.3** | **0.6** | **0.9** | **1.2** | | |
| WBC $(10^6/\mu l)$[2] | 439.78[a] | 463.97[b] | 467.85[b] | 471.88[d] | 469.81[d] | 3.62 | 0.0001 |
| RBC $(10^6/\mu l)$[3] | 2.42 | 2.33 | 2.25 | 2.43 | 2.43 | 0.07 | 0.3591 |
| Hb (g/dl)[4] | 8.55 | 8.49 | 8.46 | 8.57 | 8.52 | 0.08 | 0.6181 |
| HCT (%)[5] | 37.45[a] | 43.19[b] | 45.33[c] | 45.67[c] | 45.57[c] | 1.77 | < .0001 |
| MCHC (g/dl)[6] | 22.57[a] | 24.08[b] | 26.22[c] | 27.46[d] | 26.40[c] | 0.29 | 0.0018 |
| ALP (IU/ml)[7] | 23.13[d] | 20.83[c] | 19.73[b] | 17.43[a] | 19.93[b] | 0.38 | 0.0004 |
| ALT (IU/ml)[8] | 42.44[e] | 38.00[d] | 37.01[c] | 30.77[a] | 33.40[b] | 0.70 | 0.0002 |
| AST (IU/ml)[9] | 80.23[e] | 70.82[c] | 72.13[d] | 65.86[a] | 69.25[b] | 0.74 | < .0001 |
| Glucose (mg/dl) | 3.42 | 3.28 | 3.45 | 3.49 | 3.50 | 0.06 | 0.1163 |

[1]Selenium levels = basal diet containing supplementation of selenium methionine @ 0 mg/kg (0.0), 0.3 mg/kg (0.3), 0.6 mg/kg (0.6), 0.9 mg/kg, (0.9) and 1.2 mg/kg (1.2);

[2]WBC = White blood cells

[3]RBC = Red blood cells

[4]Hb = Hemoglobin

[5]HCT = Hematocrit

[6]MCHC = Met hematocrit

[7]ALP = Alanine phosphatase

[8]ALT = Alanine transaminase

[9]AST = Aspartate transaminase.

[a-e] superscripts within the same row differ significantly at P<0.05 while, cells within the same row containing shared subscript have no statistically significant difference (P>0.05).

**Table 7. TBARS and antioxidant status of juvenile silver carp fed different levels of selenium methionine.**

| Parameters | Selenium levels[1] | | | | | SEM | *P-linear* |
|---|---|---|---|---|---|---|---|
| | **0.0** | **0.3** | **0.6** | **0.9** | **1.2** | | |
| **TBARS (μg/mg)** | | | | | | | |
| Serum | 3.47 | 3.45 | 3.51 | 3.56 | 3.42 | 0.04 | 0.7351 |
| Muscle | 6.56 | 6.66 | 6.80 | 6.40 | 6.62 | 0.08 | 0.4084 |
| **Superoxide dismutase (μ/mg)** | | | | | | | |
| Liver | 6.58[a] | 6.87[b] | 7.13[c] | 7.82[d] | 7.77[d] | 0.12 | 0.0265 |
| Whole body | 6.46 | 6.74 | 6.71 | 6.60 | 6.62 | 0.09 | 0.1263 |
| **Catalases (μ/mg)** | | | | | | | |
| Muscle | 72.57[a] | 73.26[b] | 73.33[b] | 73.35[b] | 73.49[b] | 0.27 | 0.0211 |
| Liver | 75.12[a] | 77.04[b] | 76.94[b] | 77.78[b] | 79.47[c] | 0.45 | 0.0067 |
| Whole body | 73.33[a] | 75.37[b] | 75.50[b] | 75.43[b] | 75.50[b] | 0.11 | < .0001 |
| **Glutathione peroxidase (μ/mg)** | | | | | | | |
| Liver | 234.20[a] | 286.50[c] | 281.40[b] | 392.70[d] | 395.30[d] | 12.89 | 0.0006 |
| Whole Body | 225.30[a] | 272.90[b] | 274.40[c] | 276.20[d] | 276.10[d] | 2.45 | < .0001 |

[1]Selenium levels = basal diet containing supplementation of selenium methionine @ 0 mg/kg (0.0), 0.3 mg/kg (0.3), 0.6 mg/kg (0.6), 0.9 mg/kg, (0.9) and 1.2 mg/kg (1.2).
[a-e] superscripts within the same row differ significantly at P<0 .05 while, cells within the same row containing shared subscript have no statistically significant difference (P>0.05).

requirement of silver carp. The Se bioaccumulation in the liver and kidney was significantly increased in this experiment and the results were consistent with previous research [22, 23, 27]. This higher Se deposition in tissues enhances the selenoproteins formation which results in better antioxidant capacity of the fish to combat the stress [28]. It is well documented that greater selenoprotein expression during oxidative stress demands more Se availability [29]. The changes in tissue Se concentration are closely related to fish health [30]. In this experiment, the results of the proximate analysis showed no changes in fish chemical composition, indicating that the silver carp meat composition is less sensitive to dietary Se supplementation. These results are in agreement with Mushtaq, [1] for silver carp, Zhou, [31] for crucian carp, and Le and Fotedar, [32] for juvenile yellowtail kingfish. Selenium conjugates with cysteine amino acid and forms selenocysteine which is an active part of the glutathione peroxidase system in this way, Se plays a vital role to strengthen the antioxidant defense system. During oxidative stress, enzymes like SOD, CAT, and GPx play scavenging effects on ROS [7]. In our experiment, the activities of GPx, SOD, and CAT were significantly increased in the supplemented groups and were highest in 0.9 and 1.2 levels of Se than in other levels. This data suggests that supplementation below 0.9 mg Se/kg is less effective to combat oxidative stress. Similar results regarding GPx, CAT, and SOD activities in liver tissues of loach were reported by Hao, [33]. In addition, several studies on different fish species evaluated different Se sources and mostly reported that Se supplementation strengthens the antioxidant system by improving GPx activities [12, 28]. However, TBARS values were not changed across the treatments in our experiment. TBARS test is used to measure the degree of lipid peroxidation and its lower index can be related to better shelf life as well as the quality of the meat [34]. The digestive and metabolic products of dietary ingredients are directly absorbed in the blood which resulted alters the blood's biochemical properties. In the current experiment, we found a higher concentration of WBCs, HCT, and MCHC for 0.9 and 1.2 Se levels fed groups than in the rest of the treatments. It is well established that WBCs concentration has a direct relation with lysozyme activities [35]. In addition, activities of ALT, AST, and ALP were lowered in the 0.9 mg

Se supplemented group compared with the rest of the treatments which suggests better liver health, and fish antioxidant status [36]. These results are in agreement with Abdel-Tawwab, [37] as they also documented higher AST and ALT activities in Se-fed African catfish.

## Conclusion

Considering current experiment results, dietary supplementation with selenomethionine improved the feed intake, growth performance, and feed efficiency of intensively reared juvenile silver carp. In addition, it increased the bioaccumulation of selenium in body organs and strengthen the oxidative burst. The Se methionine 0.9 mg kg$^{-1}$ dietary inclusion resulted in increased concentrations of GPX, SOD, and CAT which are involved in the activation of the antioxidant defense system. The oxidative stress-related hematological parameters like WBCs, HCT, and MCHC were greater and liver health indicators (ALT, AST, and ALP) were improved by feeding 0.9 mg kg$^{-1}$ dietary Se methionine. However, meat composition parameters such as dry matter, crude protein, and crude fat contents were not improved through dietary Se supplementation. The broken line regression analyses were suggesting to 0.9 mg kg$^{-1}$ an optimum dietary level of Se methionine for juvenile silver carp.

## Supporting information

**S1 Raw data.**
(XLSX)

## Author Contributions

**Conceptualization:** Maida Mushtaq, Mahroze Fatima, Noor Khan, Saima Naveed.

**Data curation:** Maida Mushtaq, Muhammad Khan.

**Formal analysis:** Maida Mushtaq, Muhammad Khan.

**Investigation:** Mahroze Fatima.

**Methodology:** Maida Mushtaq, Mahroze Fatima.

**Resources:** Mahroze Fatima, Noor Khan, Saima Naveed.

**Software:** Muhammad Khan.

**Supervision:** Mahroze Fatima, Syed Zakir Hussain Shah, Noor Khan, Saima Naveed.

**Validation:** Muhammad Khan.

**Visualization:** Muhammad Khan.

**Writing – original draft:** Maida Mushtaq, Muhammad Khan.

**Writing – review & editing:** Muhammad Khan.

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
