## [Decision Letter · Decision Letter 0]

19 Aug 2022

PONE-D-22-20867Optimization of Organic Selenium Requirement for Intensively Reared Hypophthalmichthys molitrix by Using Broken Line Regression on Growth PerformancePLOS ONE Dear Dr. Maida Mushtaq,

Thank you for submitting your manuscript to PLOS ONE. After careful consideration, we feel that it has merit but does not fully meet PLOS ONE’s publication criteria as it currently stands. Therefore, we invite you to submit a revised version of the manuscript that addresses the points raised during the review process.

We look forward to receiving your revised manuscript.

Kind regards,

Muhammad A.B. Siddik, Ph.D

Academic Editor

PLOS ONE

Journal Requirements:

3. Please ensure that you refer to Figure 1 in your text as, if accepted, production will need this reference to link the reader to the figure.

Additional Editor Comments:

The work is interesting. Relevant results have been obtained with a perspective for implementation in practice. I have some observation to the authors.

1. The toxic effects of selenium should be discussed. As well as the effects of insufficient selenium intake by the population should be discussed. https://doi.org/10.3390/antiox11081572

2. Author should provide a justification of choosing selenium levels of 0 mg/kg (0.0),0.3 mg/kg (0.3), 0.6 mg/kg (0.6), 0.9 mg/kg, (0.9) and 1.2 mg/kg.

3. The conclusion should be written in more detail and reflect the results obtained in the study.

Reviewers' comments:

Reviewer's Responses to Questions

**Comments to the Author**

1. Is the manuscript technically sound, and do the data support the conclusions?

Reviewer #1: Yes

2. Has the statistical analysis been performed appropriately and rigorously? 

Reviewer #1: Yes

3. Have the authors made all data underlying the findings in their manuscript fully available?

Reviewer #1: Yes

4. Is the manuscript presented in an intelligible fashion and written in standard English?

Reviewer #1: Yes

5. Review Comments to the Author

Reviewer #1: Comment #1: Abstract should be more informative

Comment #2: Revise the English writing

Comment #3: The introduction section suggested to summarized

Comment #4: In case of data in tables, decimal should be the same

Comment #5: The keywords should have different words from the title

Comment #6: The authors should update the list of references to contain recent publications in 2020 and 2021

Comment #7: The manuscript needs thorough English proofreading for grammatical mistakes and run-on sentences.

Comment #8: When write fish name for the first time, add scientific name along with common name.

Comment #8: title suggested to be revised

Comment #9: Line 28: revise the written

Comment #10 provide weight as mean = - S.D

Comment #11: in abstract results must be written with statistical view and P value should be provided

Comment #11: better conclusion must be proved at the end of abstract

Comment #12-line 90 , provide reference

Comment #15: Line 33: average fish weight ± SD or SE

Comment #16: Line 55-61: This part is unnecessary, the authors should consider to revise it

Comment # 17: The up to date references would make the discussion part better.

Comment #18: The authors should follow the journal format

6. PLOS authors have the option to publish the peer review history of their article (what does this mean?). If published, this will include your full peer review and any attached files.

Reviewer #1: **Yes: **Ehab El-Haroun

---

## [Author Response · Author response to Decision Letter 0]

31 Aug 2022

Revised and the manuscript is updated by following the given guidelines to meet PLOS ONE's style requirements.

Revised and updated please see the track changes in materials and methods.

3. Please ensure that you refer to Figure 1 in your text as, if accepted, production will need this reference to link the reader to the figure.

Revised and updated please see the track changes in growth performance results.

Additional Editor Comments:

The work is interesting. Relevant results have been obtained with a perspective for implementation in practice. I have some observations to the authors.

1. The toxic effects of selenium should be discussed. As well as the effects of insufficient selenium intake by the population should be discussed. https://doi.org/10.3390/antiox11081572

The relevant information regarding the effects of Se overdosage, and deficiency is included in the introduction section please see the tract changes in the second paragraph.

2. Author should provide a justification of choosing selenium levels of 0 mg/kg (0.0),0.3 mg/kg (0.3), 0.6 mg/kg (0.6), 0.9 mg/kg, (0.9) and 1.2 mg/kg.

This work is from a Ph.D. thesis that contained a series of experiments. In the first experiment, published in aquaculture reports, as detailed in the introduction we evaluated three Se sources () and two levels (0.5 and 1 mg/kg of the diet). As results predicted Se methionine as the best source, we aimed this experiment to narrow down the requirements by selecting 0 mg/kg (0.0),0.3 mg/kg (0.3), 0.6 mg/kg (0.6), 0.9 mg/kg, (0.9) and 1.2 mg/kg levels. Moreover, literature also in support of these dietary levels as similar levels of nano-selenium was investigated in grass carp Ctenopharyngodon idella fed with a high-fat diet. Here is a link for detail https://doi.org/10.1111/anu.13016

3. The conclusion should be written in more detail and reflect the results obtained in the study.

Revised and updated please see the track changes in the conclusion section.

Reviewers' comments:

Reviewer's Responses to Questions

Comments to the Author

1. Is the manuscript technically sound, and do the data support the conclusions?

Reviewer #1: Yes

2. Has the statistical analysis been performed appropriately and rigorously?

Reviewer #1: Yes

3. Have the authors made all data underlying the findings in their manuscript fully available?

Reviewer #1: Yes

4. Is the manuscript presented in an intelligible fashion and written in standard English?

Reviewer #1: Yes

5. Review Comments to the Author

Reviewer #1: 

Comment #1: Abstract should be more informative

Revised according to required information please find the track changes in the abstract section 

Comment #2: Revise the English writing

Revised and Grammarly software was used to detect grammar mistakes. 

Comment #3: The introduction section suggested to summarized

Revised, updated, and summarized. Please see the track changes.

Comment #4: In case of data in tables, decimal should be the same

Revised, and updated, Please check all the tables.

Comment #5: The keywords should have different words from the title

Thank you for enhancing our knowledge about it. We revised and updated please according to your worthy suggestions please find. 

Comment #6: The authors should update the list of references to contain recent publications in 2020 and 2021

Comment #7: The manuscript needs thorough English proofreading for grammatical mistakes and run-on sentences.

Revised and Grammarly software was used to detect grammar mistakes. 

Comment #8: When write fish name for the first time, add scientific name along with common name.

Please check it at the last of the abstract section line 14-15.

Comment #8: title suggested to be revised

Revised please find the track changes 

Comment #9: Line 28: revise the written

Revised please find the track changes 

Comment #10 provide weight as mean = - S.D

Initial body weight and lengths are updated in both abstract as well as material method mean with S.D. Please check lines 16, 81-82. 

Comment #11: in abstract results must be written with statistical view and P value should be provided

Revised according to required information please find the track changes in the abstract section 

Comment #11: better conclusion must be proved at the end of abstract

Revised according to required information please find the track changes in the abstract section 

Comment #12-line 90 , provide reference

Comment #15: Line 33: average fish weight ± SD or SE

Revised and corrected ± SD please find the track changes 

Comment #16: Line 55-61: This part is unnecessary, the authors should consider to revise it

Revised and updated please check the track changes.

Comment # 17: The up to date references would make the discussion part better.

Updated and some more recent references are included in the introduction and discussion sections. Please see the track changes.

Comment #18: The authors should follow the journal format.

Revised and updated please check it.

---

## [Editor Report · Decision Letter 1]

5 Sep 2022

Evaluation of dietary selenium methionine levels and their effects on growth performance, antioxidant status, and meat quality of intensively reared juvenile Hypophthalmichthys molitrix

PONE-D-22-20867R1

Dear Dr. Maida Mushtaq,

We’re pleased to inform you that your manuscript has been judged scientifically suitable for publication and will be formally accepted for publication once it meets all outstanding technical requirements.

Kind regards,

Muhammad A.B. Siddik, Ph.D

Academic Editor

PLOS ONE
---

## [Editor Report · Acceptance letter]

7 Sep 2022

PONE-D-22-20867R1 

Evaluation of dietary selenium methionine levels and their effects on growth performance, antioxidant status, and meat quality of intensively reared juvenile *Hypophthalmichthys molitrix*

Dear Dr. Mushtaq:

I'm pleased to inform you that your manuscript has been deemed suitable for publication in PLOS ONE. Congratulations! Your manuscript is now with our production department. 

Kind regards, 

on behalf of

Dr. Muhammad A.B. Siddik 

Academic Editor

PLOS ONE